# Partial Transfer Ensemble Learning Framework: A Method for Intelligent Diagnosis of Rotating Machinery Based on an Incomplete Source Domain

**DOI:** 10.3390/s22072579

**Published:** 2022-03-28

**Authors:** Gang Mao, Zhongzheng Zhang, Sixiang Jia, Khandaker Noman, Yongbo Li

**Affiliations:** MIIT Key Laboratory of Dynamics and Control of Complex System, School of Aeronautics, Northwestern Polytechnical University, Xi’an 710072, China; mg0207@yeah.net (G.M.); zhangzhongzheng@mail.nwpu.edu.cn (Z.Z.); sixiang_j@163.com (S.J.); khandakernoman93@nwpu.edu.cn (K.N.)

**Keywords:** partial transfer learning, ensemble strategy, fault diagnosis, deep adversarial convolutional neural network

## Abstract

Most cross-domain intelligent diagnosis approaches presume that the health states in training datasets are consistent with those in testing. However, it is usually difficult and expensive to collect samples under all failure states during the training stage in actual engineering; this causes the training dataset to be incomplete. These existing methods may not be favorably implemented with an incomplete training dataset. To address this problem, a novel deep-learning-based model called partial transfer ensemble learning framework (PT-ELF) is proposed in this paper. The major procedures of this study consist of three steps. First, the missing health states in the training dataset are supplemented by another dataset. Second, since the training dataset is drawn from two different distributions, a partial transfer mechanism is explored to train a weak global classifier and two partial domain adaptation classifiers. Third, a particular ensemble strategy combines these classifiers with different classification ranges and capabilities to obtain the final diagnosis result. Two case studies are used to validate our method. Results indicate that our method can provide robust diagnosis results based on an incomplete source domain under variable working conditions.

## 1. Introduction

Rotating components play a significant role in system performance and are widely applied in engineering machinery such as aerobat, engine, and gearbox systems [1,2]. The failure of rotating components may cause unexpected downtime and economic losses. Therefore, it is crucial to precisely identify and detect the fault states of rotating machinery [3]. Recently, intelligent fault diagnosis has become a hotspot because it can analyze vast amounts of measured data and provide intuitionistic diagnosis results [4].

Intelligent fault diagnosis has received a lot of attention in recent years from both industrial engineers and academic researchers and has accomplished remarkable achievements [5]. For example, shallow machine learning techniques such as support vector machine (SVM) [6] and random forest (RF) [7] have been studied. Deep learning methods have been researched that can adaptively extract the fault features hidden in a collected signal, such as recurrent neural network (RNN) [8], convolutional neural network (CNN) [9], and stack autoencoder (SAE) [10]. In addition, some variant models are being studied, such as dilated CNN [11], CNN with capsule network [12], and multiscale CNN [13]. However, the existing methods are developed based on statistics, which assume that adequate labeled samples are obtainable to train the models. In addition, these methods require the data distribution of training and testing to be identical [14]. In actual industry settings, obtaining a large amount of labeled data is unrealistic. Even if the labeled data can be acquired, the aforementioned methods may fail to recognize the unlabeled data collected from another machine or under different working conditions due to the inconsistent data distribution [15].

The proposal of transfer learning aims to solve this problem by promoting models trained by labeled data from a relevant domain to the target fields [16]. The implementation of transfer learning for machine fault diagnosis mainly includes two scenarios: (1) A few target-domain-labeled data are available but are insufficient to support the model training. Qian et al. [17] implemented bearing fault diagnosis under diverse working conditions by transferring the parameters of SAE. Chen et al. [18] studied the use of transferable CNN to recognize the fault states of rotary machinery by pre-training a 1D-CNN using the source data and fine-tuning it with the limited labeled samples in the target domain. (2) There are no available labeled target data to participate in the model training process. One solution is to add a domain adaptation term to the loss function, such as the Maximum Mean Discrepancy (MMD) [4,19,20], Wasserstein distance [21]. Another solution is to implement the transfer learning by use of an adversarial network, in which case a feature extractor aims to extract domain-insensitive features from the target and source domains by adversarial training [22,23,24].

The existing cross-domain fault diagnosis methods can obtain superior results in the target domain, but the precondition lies in the assumption that the health states in the target domain are identifiable with the source domain. However, given the variation of operations and unpredictability of the fault states, it is difficult to guarantee that the current or future fault states have all been learned in the training phase. Therefore, the source training dataset is usually incomplete, and there are some additional failure states in the target domain. This causes negative transfer and misclassification in the testing stage. These private failure state data can be collected from another component, but the working conditions, such as speed, load, and frequency, are completely different from the source domain and target test data. Figure 1 shows an example of such a situation. Dataset *A* is collected from bearing 1 and contains five health states. However, during the test, more fault states appeared due to the change in working conditions, resulting in seven health states. The data for the two missing health states can be supplemented from dataset *B*. Dataset *B* is collected from bearing 2 and includes four health states total. So, the data source domain discrepancy between *A* and *B* also needs to be taken into consideration; this creates some difficulties for the implementation of transfer learning diagnostic methods.

This research studies a partial transfer ensemble learning framework (PT-ELF) to solve the above problem. First, two incomplete source domain datasets collected from different components or under different working conditions are defined. Note that neither of them contains all the health states present in the target domain data. They are used to form a complete dataset in which all the health states are included. Then, a weak global classifier based on the complete dataset and two partially strong classifiers based on the deep adversarial network are established. Finally, since the classification ability and classification range of classifiers differ, a particular ensemble strategy is designed to combine these two strong partial classifiers and the weak global classifier, resulting in the final diagnostic results. The main contributions of this research are summarized as follows:(1)A partial transfer ensemble learning framework is designed to diagnose the fault with incomplete training datasets under various conditions;(2)To incorporate the classification ability of multiple classifiers into the PT-ELF model, a particular ensemble strategy is designed to combine a weak global classifier and two partial domain adaptation classifiers;(3)Two case studies using rotor bearing test bench data and motor bearing data are performed to validate and demonstrate the superiority of the proposed method.

The rest of this article is arranged as follows: Section 2 presents the basic theories. The details of the proposed PT-ELF are given in Section 3. Section 4 validates the proposed method and analyzes the results. Finally, the conclusion in Section 5 brings the study to a close.

## 2. Basic Theory

### 2.1. Convolutional Neural Network

A standard CNN usually includes convolution, pooling, fully connected, and output layers. In addition, batch normalization operation is usually used in CNN [25]. A convolution layer is combined with a pooling layer to form a convolution block, and a deep architecture is built from several such blocks. A Softmax Regression layer usually serves as the last layer and performs regression or classification [26]. In a convolutional layer, the local receptive is adopted, in which only part of the input sample points connect to each node. This operation rapidly decreases the number of parameters and the model complexity. To identify the local features throughout the input sample, weights and biases are shared between the hidden neurons in one convolutional layer [27]. The process in the convolutional layer can be expressed as:(1)$$z_{n}^{l}=\sum_{k}x_{k}^{l-1}*w_{n}^{l}+b_{n}^{l}$$
where $x_{k}^{l-1}$ is the *k*-th node in *l* − 1 layer. * represents the convolution operation. $w_{n}^{l}$ and $b_{n}^{l}$ represent the weight and the corresponding bias. Additionally, the activation function φ(•) is given to transform the convolution layers nonlinearly, which can be denoted as:(2)$$c_{n}^{l}=\varphi (z_{n}^{l})$$
where $c_{n}^{l}$ represents the *k*-th nonlinear feature value in *l* − 1 layer. Sigmoid and ReLU activation functions are commonly used in CNN. Sigmoid can normalize the input data to between 0 and 1. ReLU can enhance the efficiency of the model training and decrease the risk of gradient disappearance [28].

In a pooling layer, the down-sampling operation can decrease the dimension of the features and enhance their robustness. Mathematically, a maximum pooling operation is defined as:(3)$$po_{j}=\underset{i\in m_{j}}{\max\{c_{j}(i)\}}$$
where $c_{j}$ represents the *j*-th location, and the $po_{j}$ is the output of the pooling. For classification tasks, after several convolution blocks and fully connected layers, the Softmax function is usually utilized to predict categories. The loss objective function can be expressed as:(4)$$H(r,p)=-\sum_{i}r_{i}log(p_{i})$$
where *p* represents the output probability, and *r* corresponds to the actual labels.

### 2.2. Deep Adversarial Convolutional Neural Network

Generally, a deep adversarial convolutional neural network (DACNN) consists of a feature extractor *G_f_*, a domain discriminator *G_d_*, and a classifier *G_y_* [29,30,31]. The feature extractor, namely several convolution blocks, serves as a contestant in the DACNN. It can be expressed as $G_{f}=G_{f}(x,\;\theta_{f})$, which indicates that the features are extracted from the input sample *x* with parameters $\theta_{f}$. In addition, a discriminator (binary classifier) is treated as the opponent, which is expressed as $G_{d}=G_{d}(G_{f}(x),\;\theta_{d})$. Input the source and target samples into the feature extractor, and the output features are further distinguished by the discriminator *G_d_*. The binary cross-entropy loss is taken as an objective function, which is described as:(5)$$L(G_{d}(G_{f}(x_{i})),d_{i})=d_{i}\log\frac{1}{G_{d}(G_{f}(x_{i}))}+(1-d_{i})\times \log\frac{1}{1-G_{d}(G_{f}(x_{i}))}$$
where *d_i_* denotes the binary variable for *x_i_*. Through the adversarial training between two parts, the feature extractor *G_f_* tends to extract the common features from the two types of data and makes it hard to differentiate 0 or 1 as the discriminator. Hence, the model can perform well on both the source and target datasets. The loss function is expressed as:(6)$$E(\theta_{f},\;\theta_{d})=-(\frac{1}{n}\sum_{i=1}^{n}L_{d}^{i}(\theta_{f},\;\theta_{d})+\frac{1}{N-n}\sum_{i=n+1}^{N}L_{d}^{i}(\theta_{f},\;\theta_{d}))$$
where *n* and *N* − *n* represent the sample number of the source and target domain.

Additionally, all of the labeled samples should be supervised during training to ensure the accuracy of the diagnosis in the adversarial procedure. Thus, a classifier is established and is expressed as $G_{y}=G_{y}(G_{f}(x),\;\theta_{y}):R^{D}\to R^{L}$ with parameters $\theta_{y}$, in which *L* is the number of classes. The cross-entropy loss is applied in the Softmax function and is described as:(7)$$L(G_{y}(G_{f}(x_{i})),y_{i})=\log\frac{1}{G_{y}{\left(G_{f}(x_{i})\right)}_{y_{i}}}$$

Adding Equation (7) to the objective function (6), the optimization objective can be expressed as:(8)$$E(\theta_{f},\;\theta_{y},\;\theta_{d})=\frac{1}{n}\sum_{i=1}^{n}L_{y}^{i}(\theta_{f},\;\theta_{y})-\lambda (\frac{1}{n}\sum_{i=1}^{n}L_{d}^{i}(\theta_{f},\;\theta_{d})+\frac{1}{N-n}\sum_{i=n+1}^{N}L_{d}^{i}(\theta_{f},\;\theta_{d}))$$
where $L_{y}^{i}(\theta_{f},\;\theta_{y})=L(G_{y}(G_{f}(x_{i})),y_{i})$ and λ is a non-negative hype-parameter trade-off for the losses of the discriminator. In the whole training procedure of the DACNN, the optimization parameters $\theta_{f}$, $\theta_{y}$, $\theta_{d}$ can be obtained by:(9)$$\left(\hat{\theta_{f}},\;\hat{\theta_{y}})=\underset{\theta_{f},\;\theta_{y}}{\mathrm{argmax}}E(\theta_{f},\;\theta_{y},\;\hat{\theta_{d}}\right)$$
(10)$$\hat{\theta_{d}}=\underset{\theta_{d}}{\mathrm{argmax}}E(\hat{\theta_{f}},\;\hat{\theta_{y}},\;\theta_{d})$$

The flowchart of the DACNN is displayed in Figure 2. By optimizing Equations (9) and (10), the DACNN tends to train a feature extractor *G_f_* that can extract suitable representations from input samples that can be classified accurately by the classifier *G_y_* but weakens the ability of the discriminator *G_d_* to differentiate which domain this representation is from. In the phases of testing, the domain-insensitive features are extracted by the feature extractor G*_f_* and fed into the health state classifier *G_y_* to identify the states immediately.

## 3. The Proposed Method

This section describes the proposed method in detail. It mainly includes problem formulation, the training of the three classifiers, and the classifiers’ ensemble.

### 3.1. Problem Formulation

Before implementing the proposed method, two incomplete source domain datasets *A* and *B* are defined as shown in Figure 3. The source dataset $A=\{(x_{i}^{A},y_{i}^{A}){\}}_{i=1}^{n_{A}}$ of *n_A_* labels instances associated with |*D_A_*| classes and is drawn from distribution ***P****_SA_*. The source dataset $B=\{(x_{i}^{B},y_{i}^{B}){\}}_{i=1}^{n_{B}}$ of *n_B_* labels instances associated with |*D_B_*| classes collected from another same-type component and is drawn from distribution ***P****_SB_*. The class label spaces of *A* and *B* are denoted as *D_A_* and *D_B_*, respectively. The collection of different components results in variations in the operating conditions (such as load, speed, etc.) in a real industrial environment; this means that ***P****_SA_* ≠ ***P****_SB_*. In addition, there must be some shared health states contained in both source dataset *A* and source dataset *B*, which are denoted as $D=D_{A}\cap D_{B}$ and shown in Figure 3. ${\hat{D}}_{A}=D_{A}\backslash D_{B}$ denotes the private label space of the *A* and ${\hat{D}}_{B}=D_{B}\backslash D_{A}$ denotes the private label sets of *B*.

However, in the testing stage of the actual machine fault diagnosis scenario, all possible health states may appear. Therefore, the target domain dataset includes all health states; it can be expressed as $T=\{(x_{i}^{T}){\}}_{i=1}^{n_{T}}$ of *n_T_* unlabeled instances associated with |*D_T_*| classes drawn from distribution ***P****_T_*. The *D_T_* represents the label sets of the target domain and $D_{T}=D_{A}\cup D_{B}$. In addition, the target domain distribution ***P****_T_* is different in source domain distributions ***P****_SA_* and ***P****_SB_*.

This paper aims to establish a fault diagnosis model to realize fault diagnosis based on incomplete source training data under different operating conditions.

### 3.2. Classifier Training

This section describes the training procedure for the three classifiers (weak classifier C*_W_*, classifier C*_A_*, and classifier C*_B_*) concretely.

First, a complete dataset *C* that contains all of the classes can be formed based on the incomplete source datasets *A* and *B*, as shown in Figure 4. In the complete dataset *C*, the sample in label space ${\hat{D}}_{A}$ is from source dataset *A*, and the sample in label space ${\hat{D}}_{B}$ is from source dataset *B*. For the samples in shared label space *D*, a portion of them come from *A*, and the rest come from *B*. Thus, the label space of dataset *C* is the same as *T*, and it includes |*D_T_*| health states. Second, a standard CNN classifier C*_W_* is trained using the complete dataset *C*. However, since the source domain datasets *A* and *B* are collected under various work conditions, the samples in the dataset *C* are drawn from two types of distributions. In addition, the data distribution in the testing set ***P****_T_* is different in ***P****_SA_* and in ***P****_SB_*. Therefore, the classifier C*_W_* has poor classification ability for the target domain data. However, the classifier C*_W_* has the ability to classify all health states.

After the weak classifier C*_W_* is obtained, the test samples from the target domain $T=\{(x_{i}^{T}){\}}_{i=1}^{n_{T}}$ of *n_T_* unlabeled instances associated with |*D_T_*| classes are classified, and the result is served as a pseudo-label to participate in the subsequent training. Target domain samples whose pseudo-label is in *D_A_* are obtained to construct the target domain training set *A*_T_. The samples whose pseudo-label is in *D_B_* are obtained to construct the target domain training set *B*_T_. Thus, the datasets *A* and *A*_T_ have the same label space *D_A_*, and the datasets *B* and *B*_T_ have the same label space *D_B_*.

Dataset *A* and *A*_T_ have the same health states but draw from different distributions. So, a DACNN model can be trained using the datasets *A* and *A*_T_. A feature extractor and a classifier in this DACNN are combined to form a block, which is taken as classifier C*_A_*. The classifier C*_A_* is constructed by a DACNN using domain adaptation techniques, so that it has a strong classification ability for the unlabeled target domain dataset. However, the classification range of strong classifier C*_A_* is limited to |*D_A_*| classes. After the training of classifier C*_A_* is completed, classifier C*_B_* is trained in the same way. Similarly, the classification range of C*_B_* is limited to |*D_B_*| classes.

In the implementation process of the DACNN, the SELU activation function is used in convolutional layers; its mathematical expression is expressed as Equation (11):(11)$$\mathrm{SELU}(x)=\lambda \{\begin{array}{ll} \alpha e^{x}-\alpha & \left(x\leq 0\right)\\ x & \left(x>0\right) \end{array}$$
where the value of *α* is 1.6732, and the value of *λ* is 1.0507. The SELU activation function can automatically normalize the sample distribution to 0 mean value and unit variance to avoid the gradient exploding or disappearing. The activation function used in the fully connected layer in the state classifier and domain discriminator is ReLU, and it is expressed as Equation (12): (12)$$\mathrm{ReLU}(x)=\{\begin{array}{ll} 0 & \left(x\leq 0\right)\\ x & \left(x>0\right) \end{array}$$

In this way, three well-trained classifiers are achieved, including one weak global classifier C*_W_*, one strong partial classifier C*_A_*, and one strong partial classifier C*_B_*. The details of the three classifiers are listed in Table 1.

### 3.3. Classifiers’ Ensemble

After the three classifiers are obtained, this section designs a particular ensemble strategy to combine their results. The procedure for the ensemble strategy is presented in Figure 5. 

After inputting a testing sample ***x*** into the three classifiers, the classification result **y***_W_*, **y***_A_*, and **y***_B_* can be output from the three classifiers, which can be expressed as:(13)$$\{\begin{array}{c} y_{W}=C_{W}(x)\\ y_{A}=C_{A}(x)\\ y_{B}=C_{B}(x) \end{array}$$

If $y_{W}=y_{A}∥y_{W}=y_{B}∥y_{A}=y_{B}$ is satisfied, the final result **y** can be obtained by a majority voting strategy immediately. Otherwise, it means that the results of the three classifiers are different from each other. In such cases, because the classifier C*_W_* is a global classifier, **y***_W_* is served as the reference standard. If **y***_W_* ∈ *D_A_* is satisfied, that means that the actual label of ***x***may be in *D_A_*. In this range, the classifier C*_A_* has perfect classification ability, and thus **y***_A_* is served as the final result. Similarly, if **y***_W_* ∈ *D_B_* is satisfied, **y**_B_ is served as the final result. However, if **y***_W_* ∈ *D* is satisfied, both the classifiers C*_A_* and C*_B_* have good classification ability in this shared range. In this case, **y** is determined according to the output probability *p* in the Softmax layer of classifiers, and it can be expressed as:(14)$$\{\begin{array}{c} y=y_{A}\;if\;p_{A}=\max(p_{A},\;p_{B},\;p_{W})\\ y=y_{B}\;if\;p_{B}=\max(p_{A},\;p_{B},\;p_{W})\\ y=y_{W}\;if\;p_{W}=\max(p_{A},\;p_{B},\;p_{W}) \end{array}$$
where the *p_A_*, *p_B_*, and *p_W_* represent the Softmax output probability of classifiers C*_A_*, C*_B_*, and C*_W_*; max(·) is the maximum function. 

### 3.4. Architecture of the Proposed Method

The architecture of our method for fault diagnosis is presented in Figure 6, and the process is summarized below. 

(1)Collect original vibration signals from different components or under different working conditions, and convert them into frequency domain signals for subsequent model training;(2)Construct a complete dataset by combing these incomplete datasets, and train a weak global classifier CNN;(3)Classify the target domain data using the weak classifier to obtain the two target domain training sets;(4)Train two DACNN models using two source datasets and target domain training sets to construct two strong partial classifiers;(5)Design a particular ensemble strategy to combine the three classifiers and obtain the final classification results.

## 4. Experimental Verification

To validate the effectiveness of the proposed PT-ELF method, rotor and rolling bearing experiments are designed. Note that the code for the proposed method is written in Pytorch 1.2 and runs with 16G RAM and a Core I5 10400F CPU.

### 4.1. Case 1

#### 4.1.1. Rotor Experiment

Case 1 adopts the rotor dataset from Northwestern Polytechnical University. As shown in Figure 7a, the experimental system is composed of a three-phase variable frequency motor, single-span rotor shafting, torque speed sensor, rolling bearing seat, shafting load plate, rubbing mounting bracket, platform bottom plate, radial loading device, coupling, system control cabinet, and fault suite. A displacement sensor is mounted on the rotor test bench to collect vertical vibration signals under a health state and six different fault states as shown in Figure 8, and the sample frequency is 10,240 Hz. Figure 7b depicts the sensor and single-span rotor shaft layout. The structural components are listed in Table 2. 

The rotor vibration data are collected under three working load conditions of 0%, 20%, and 40%. As detailed in Table 3, for each load, data from seven health states (including a health state and six fault states) are used. The data in each state are divided into 300 samples, with 80 randomly selected as tests and the remaining 220 used to train. Each sample, each consisting of 800 data points, is used to verify the method proposed in this paper. Figure 9 shows the waveform of the original displacement signal and the spectral distributions of each health state under 0% load. The left shows the spectral signal, and the right shows the corresponding spectrum. The signals have a large amplitude of around 10–30 Hz, showing relatively similar characteristics, which makes it hard to recognize the health states.

#### 4.1.2. Results and Discussion

In this case study, two incomplete source datasets are constructed, as shown in Table 4. The source dataset *A* contains five kinds of health states (states 1–5), and the source dataset *B* contains four kinds of health states (states 4–7).

First, the source domain datasets *A* and *B* are mixed to form a training set that contains all health states, which is used to train a weak classifier C*_W_*. The classifier C*_W_* has a classification ability for all of the health states (seven kinds of health states). Second, according to the classification results (the pseudo-label) of the weak classifier C*_W_* on the target domain samples, two transfer models based on a DACNN are trained. They are transferred from source domain dataset *A* and source domain dataset *B* to the target domain. Thus, two strong classifiers C*_A_* and C*_B_* are trained. Finally, after classifying a test sample by the classifiers C*_A_*, C*_B_*, and C*_W_*, three results are obtained and fused by the proposed ensemble strategy described in Section 3.3.

To demonstrate that our method is applicable to various operating conditions, five test scenarios (test scenarios A1–E1) are designed to test the proposed method. As listed in Table 5, the source domain *A*, source domain *B*, and target domain are served by the collected dataset under different loads. In source dataset *A*, only five kinds of labeled samples in states 1–5 are available. Similar to source domain *A*, in source dataset *B*, only four types of labeled samples in states 4–7 are available. The test data in the target domain contain all seven kinds of unlabeled samples in states 1–7. 

The accuracies of the three classifiers (two strong partial classifiers and a weak global classifier) and the proposed PT-ELF method in the five test scenarios are listed in Table 6, and a bar diagram is shown in Figure 10a. Note that the accuracy of C*_A_* is tested by states 1–5, and the accuracy of C*_B_* is tested using states 4–7. The result of the weak classifier C*_W_*and the ensemble result are tested using target domain test data that contain all of the health states (states 1–7). 

It can be seen from Table 6 that the two strong classifiers C*_A_* and C*_B_* have high accuracy in the corresponding classification range, with averages of 93.29% and 96.83%. On the one hand, this is because the two strong classifiers are trained by a domain adversarial network DACNN, which can extract domain-insensitive features to classify. On the other hand, they are just tested by partial health states. The result of the weak classifier C*_W_* is relatively poor, with an average accuracy of 86.52%. This is because the data of the target domain and two source domains are not uniformly distributed, leading to the decrease in classification performance. 

Out of five test scenarios, the result in scenario B1 is the highest at 95.41%; scenario C1 has the lowest accuracy at 83.75%, and the average is 90.73%. This is significantly higher than the weak classifier C*_W_*, and maintains a high classification accuracy. This is because the proposed ensemble strategy can cause the test sample to be classified by the corresponding strong classifier as far as possible. It indicates that our method can still achieve good results even under incomplete training data.

In addition, to prove the superiority of our method, relevant methods for a CNN and a DACNN, trained by source dataset *A* and source dataset *B*, respectively, are used as comparison methods (Method 1–4). The result is listed in Table 7, and a bar diagram of the various methods is shown in Figure 10b. It can be observed that the average accuracies of the CNN trained by source domains *A* and *B* are 58.87% and 55.27%, respectively. The average accuracies of the DACNN trained by source domains *A* and *B* are 64.02% and 56.79%, respectively, which are significantly higher than the accuracy of the CNN. This is because the DACNN can extract domain-insensitive features using adversarial training; this restrains the model’s performance decrease caused by a distribution discrepancy and further improves the accuracy of the model in the target domain. However, since the source domain *A* is incomplete, a model (CNN or DACNN) trained by source dataset *A* is unable to classify the testing samples whose actual label is in ${\hat{D}}_{B}$ (states 6–7). Similarly, a model (CNN or DACNN) trained by source dataset *B* is unable to classify the testing samples whose actual label is in ${\hat{D}}_{A}$ (states 1–3); therefore, the results of methods 1–4 are poor compared to our method. The average accuracy of our method is as high as 90.73%, which indicates that the proposed method has good classification ability for all health states presented in the testing dataset in the target domains.

### 4.2. Case 2

#### 4.2.1. Rolling Bearing Experiment

The rolling bearing vibration data utilized in case 2 are from Case Western Reserve University [32]. As shown in Figure 11, the setup mainly consists of a loading motor, an induction motor, and testing bearings. The vibration signals used in this case are collected by an accelerometer installed near the drive end. As listed in Table 8, the vibration signals were collected under four different loads (Load 1–Load 4). Each fault was artificially implanted into the bearings with different severity levels from 0.007 to 0.028 inches in diameter (1 inch = 25.4 mm). The details of the test bearing are listed in Table 9.

The vibration data collected under four different loads are used to test the proposed method. Each of them includes 12 health states, which include different failure locations (shown in Figure 12), different failure orientations, and different failure severities. As detailed in Table 10, each health state contains 300 samples, which consist of 400 continuous data points. At random, 200 samples are selected to train, and the remaining 100 are used to test. The raw vibration is under 1797 rpm (0 hp) (in the left column), and the corresponding spectral distributions (in the right column) are shown in Figure 13. In terms of raw vibration signals, the health state vibration amplitude is relatively small (Figure 13a). The fault signals (Figure 13b–i) have an obvious impact. The spectral distribution contains the fault frequency and the bearing natural frequency. In addition to the health signals, the other fault vibration signals have a higher amplitude of around 3–4 kHz. It is still very unrealizable to accurately distinguish the fault location, dimension, and orientation across different working conditions with new fault states.

The proposed method mainly studies the case in which only partial health state labeled data are available in the source domain. To verify our method, we assume that source domain dataset *A* only contains eight kinds of fault state labeled data, while source domain dataset *B* contains seven kinds of labeled data. Among them, three categories overlap, as shown in Table 11. In addition, all target domain data are unlabeled; these data contain 12 kinds of health states.

#### 4.2.2. Results and Discussion

Similar to Case 1, the source datasets *A* and *B* are first mixed to form a training set containing all health states, and it is used to train the weak classifier C*_W_*. Thus, C*_W_* has a classification ability for all of the health states, but the classification ability is weak.

In the following step, two DACNN models are trained based on source domain datasets *A* and *B* to adapt target domain data. Then, two strong classifiers C*_A_* and C*_B_* can be obtained. In each DACNN, the feature extractor G*_f_* contains two convolution blocks. Meanwhile, the classifier G*_y_* contains a fully connected layer and output by a Softmax function. The G*_y_*(G*_f_* (*x*)) in the DACNN is taken as the classifier. Finally, three well-trained classifiers C*_A_*, C*_B_*, and C*_W_* with different classification capabilities and classification ranges are integrated using the ensemble strategy introduced in Section 3.3 to obtain the final diagnosis result.

To demonstrate that our method is applicable to different working conditions, five test scenarios (test scenarios A2–E2) with incomplete data are used to test the proposed method, as shown in Table 12. In source dataset *A*, eight kinds of labeled samples in states 1–8 are available, and in source dataset *B*, seven kinds of labeled samples in states 6–12 are available. The target data, which contains 12 kinds of unlabeled samples in states 1–12, is used to test. In the five test scenarios, source domain datasets *A* and *B* and the target domain dataset are served by data collected under different loads. To indicate the superiority of our method, two conventional deep learning methods based on CNN (method 1 and method 2) and two transfer learning methods based on DACNN (method 3 and method 4) are used for comparison in five test scenarios; the results are listed in Table 13. Method 1 and method 3 are trained using source dataset *A*, and method 2 and method 4 are trained using source dataset *B*. In order to show the comparison results visually, the results bar diagram for different methods is shown in Figure 14.

As shown in Table 13 and Figure 14, the average accuracies of methods 1 and 2 are 64.27% and 57.53%, respectively. The average accuracies of method 3 and method 4, based on transfer learning, are 66.22% and 58.05%, respectively. This is because the DACNN can solve the problem of cross-domain fault diagnosis well and enhances the recognition accuracy in the target domain. However, since the source datasets *A* and *B* are incomplete, neither of them contains all the health states presented in the testing data; the fault classification accuracy is still relatively low even if the transfer strategy is used. The accuracy of the method proposed can achieve 98.08%, 95.41%, 99.66%, 99.25%, and 95.83% in five test scenarios, respectively. Accuracy is the lowest in test scenario B2, but it can still remain at 95.41%. In test scenario C2, the classification accuracy is the highest at 99.66%. The comparison results demonstrate that the proposed PT-ELF method exhibits satisfactory cross-domain diagnostic ability with new health states.

## 5. Conclusions

This paper proposes a rotating machinery fault diagnosis method based on partial transfer learning and ensemble learning. Unlike other existing cross-domain diagnostic methods with the assumption of the same health states in the source and target domains, the proposed method can provide a reliable diagnosis result in the target domain even when the source domain is incomplete and only contains partial health states. As the core of the proposed method, partial transfer learning can avoid the problem induced by incomplete training data and train two classifiers with strong classification capabilities for partial categories. Then, a particular ensemble strategy is designed to combine the output of the three classifiers (a weak global classifier and two strong partial classifiers). The effectiveness of the proposed method is validated on a rotor experiment and a bearing experiment. After comparing with four related methods, results indicate that the proposed method can achieve superior performance and provide a reliable diagnosis result based on incomplete source domain under various working conditions.

In this preliminary study, the proposed method lies in the assumption that the missing health states in the source domain training set can be obtained from another dataset or another component. The unseen health states will be considered in our future research.

## Figures and Tables

**Figure 1 sensors-22-02579-f001:**
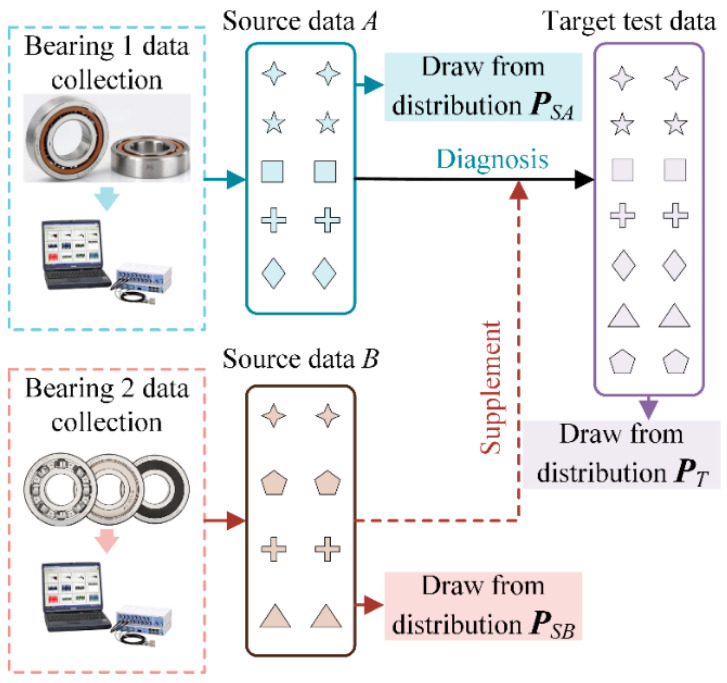
Example of the situation of fault diagnosis with new health states.

**Figure 2 sensors-22-02579-f002:**
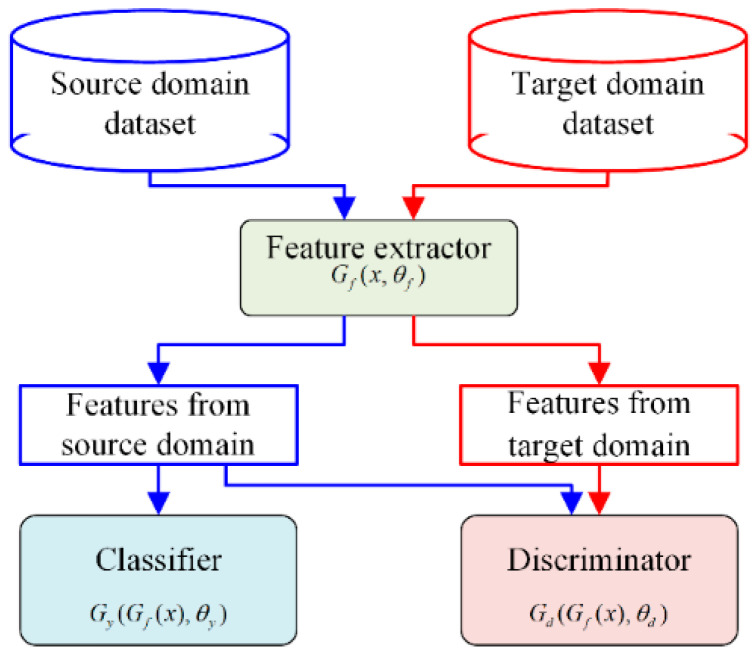
The schematic of the DACNN.

**Figure 3 sensors-22-02579-f003:**
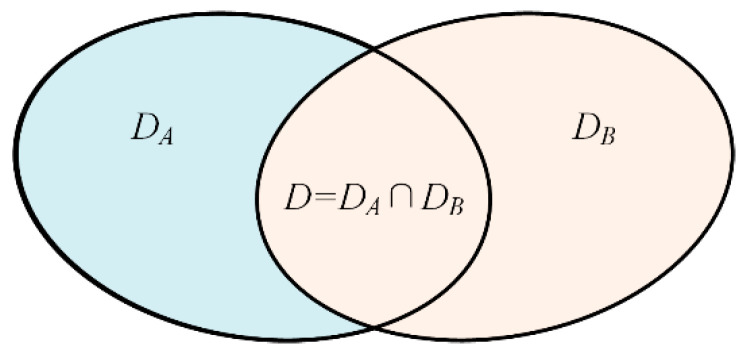
Two different source domain datasets.

**Figure 4 sensors-22-02579-f004:**
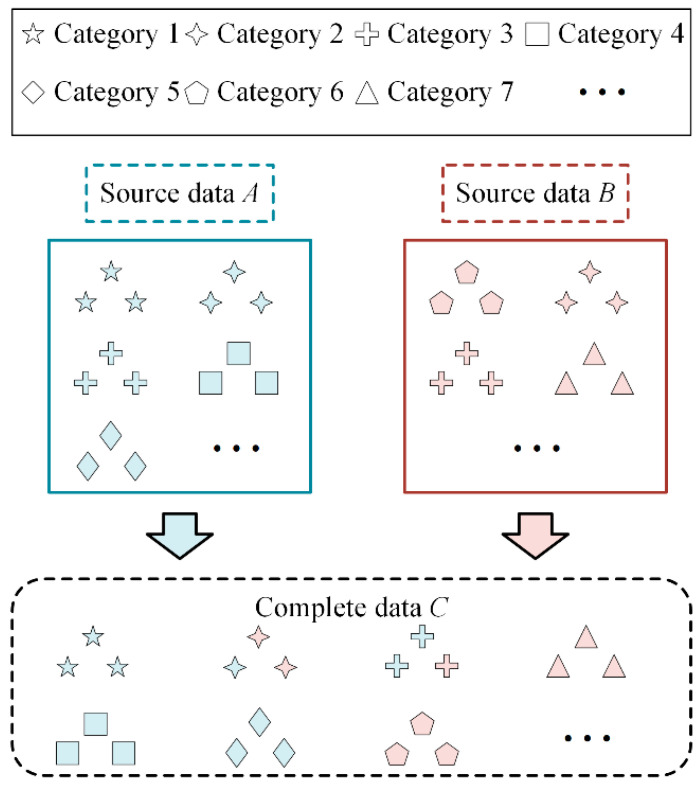
The process of forming a complete dataset C.

**Figure 5 sensors-22-02579-f005:**
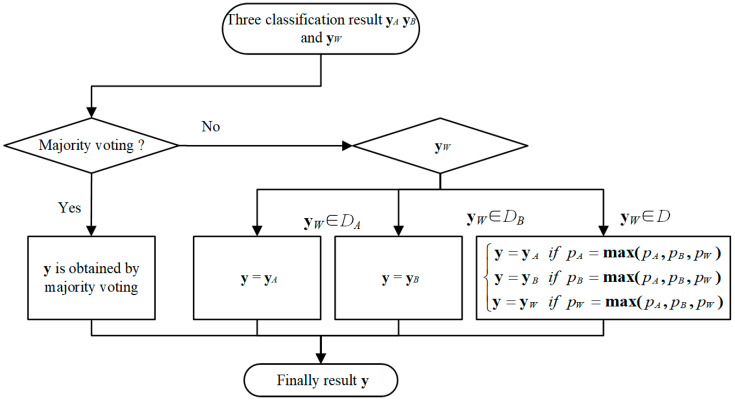
The flowchart of the classifiers’ ensemble.

**Figure 6 sensors-22-02579-f006:**
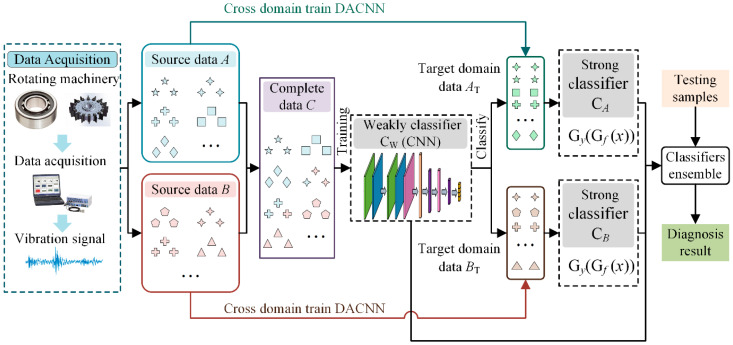
The overall procedures of the proposed method.

**Figure 7 sensors-22-02579-f007:**
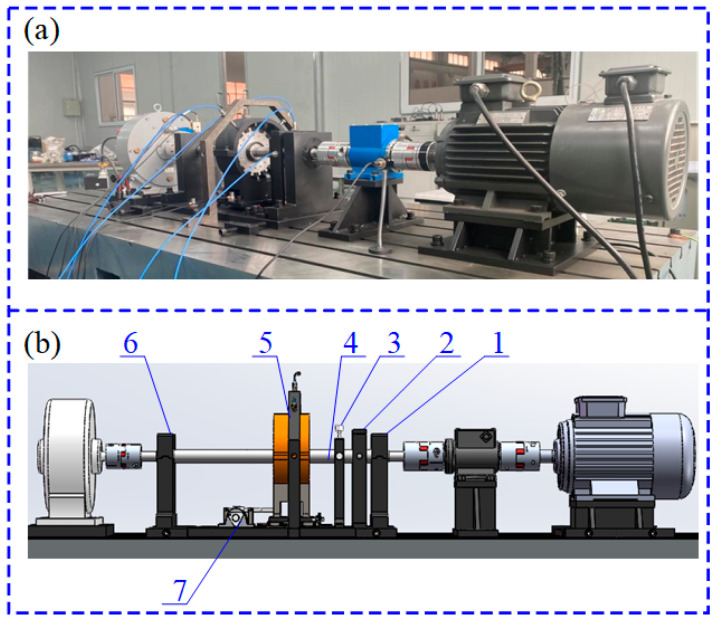
The rolling bearing experiment system: (**a**) the experimental test rig; (**b**) the layout of the test rig.

**Figure 8 sensors-22-02579-f008:**
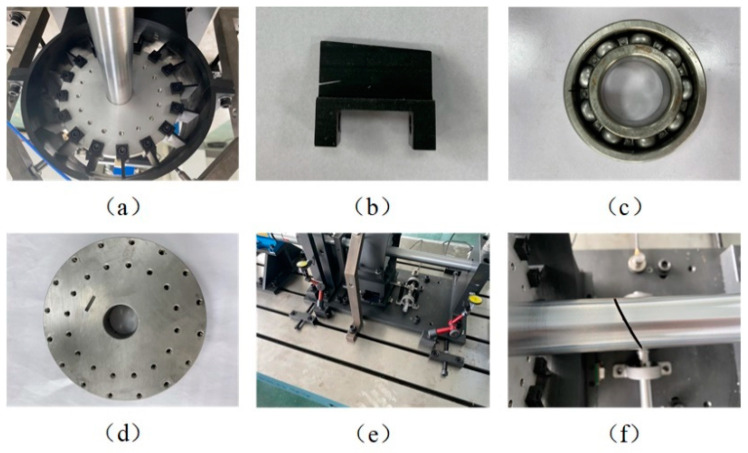
Six different fault states: (**a**) full annular rub; (**b**) blade crack; (**c**) bearing fault; (**d**) blisk crack; (**e**) Shaft coupling fault; (**f**) Shaft crack.

**Figure 9 sensors-22-02579-f009:**
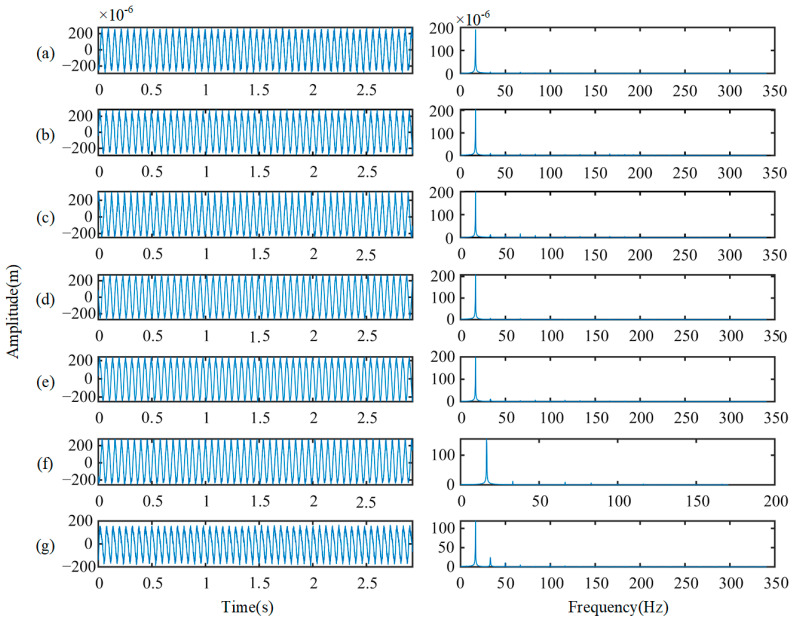
Original displacement signals and spectral distributions: (**a**) health; (**b**) full annular rub; (**c**) blade crack and bearing fault; (**d**) blade crack; (**e**) blisk crack; (**f**) shaft coupling fault; (**g**) shaft crack.

**Figure 10 sensors-22-02579-f010:**
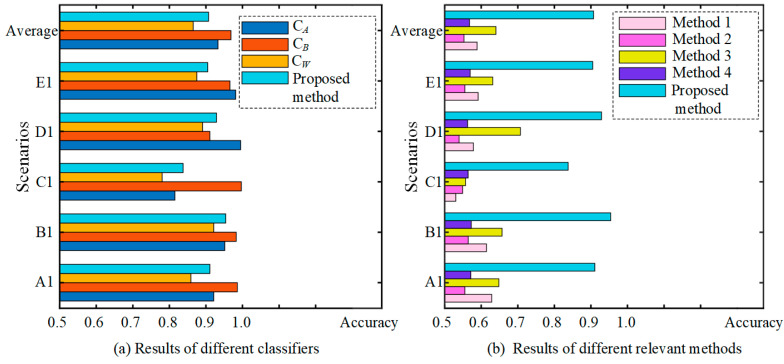
The result diagram for different classifiers.

**Figure 11 sensors-22-02579-f011:**
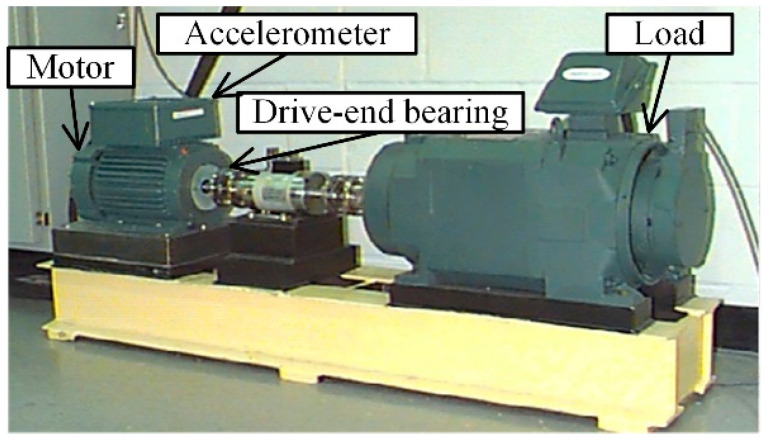
The experiment setup of rolling bearing.

**Figure 12 sensors-22-02579-f012:**
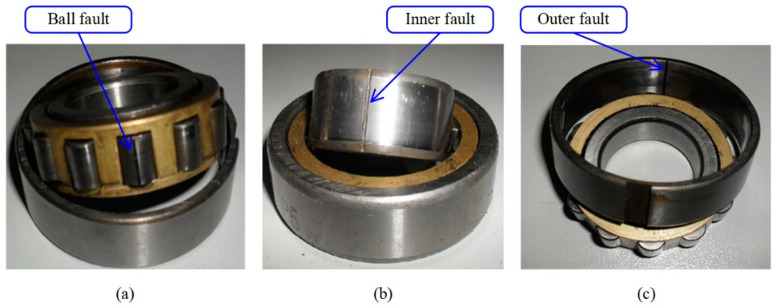
The faults of bearing in three locations: (**a**) ball fault; (**b**) inner fault; (**c**) outer fault.

**Figure 13 sensors-22-02579-f013:**
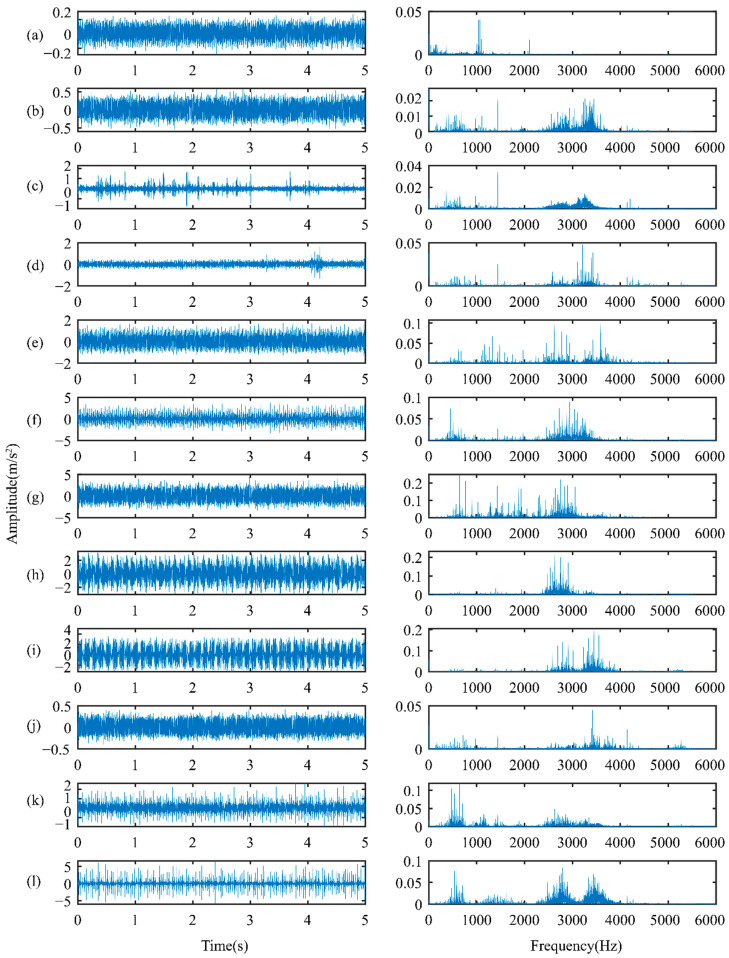
Waveform of raw signals and spectral distributions of the rolling bearing: (**a**) health; (**b**) rolling element failure (0.007); (**c**) rolling element failure (0.014); (**d**) rolling element failure (0.021); (**e**) inner race failure (0.007); (**f**) inner race failure (0.021); (**g**) inner race failure (0.028); (**h**) outer race failure (0.007 Center); (**i**) outer race failure (0.007 Vertical); (**j**) outer race failure (0.014 Center); (**k**) outer race failure (0.021 Center); (**l**) outer race failure (0.021 Vertical).

**Figure 14 sensors-22-02579-f014:**
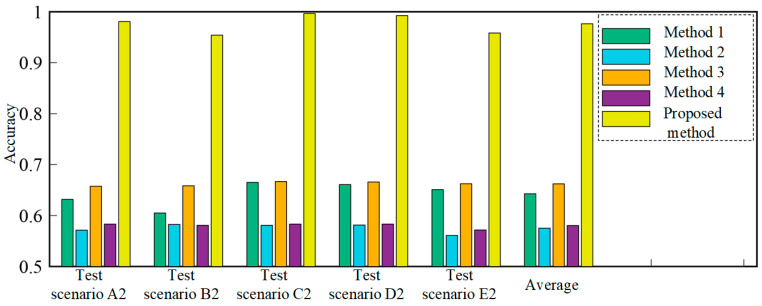
The results diagram for different methods.

**Table 1 sensors-22-02579-t001:** Classification range and ability of the three classifiers.

Classifiers	Range of Classification	Ability of Classification
C*_A_*	*D_A_* 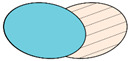	Strong
C*_B_*	*D_B_* 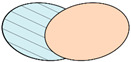	Strong
C*_W_*	*D_A_* ∪ *D_B_* 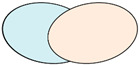	Weak

**Table 2 sensors-22-02579-t002:** The structural components of the single-span rotor shafting.

No	Component
1	Support bearing pedestal
2	Displacement sensor bracket
3	Friction assembly and bracket
4	Shaft
5	Casing friction support and blade disc
6	Test bearing pedestal
7	Worm gear and worm

**Table 3 sensors-22-02579-t003:** Seven health states of the rotor.

Label	Health States	The Number of Training/Testing Samples
0	Health	220/80
1	Full annular rub	220/80
2	Blade crack and bearing fault	220/80
3	Blade crack	220/80
4	Blisk crack	220/80
5	Shaft coupling fault	220/80
6	Shaft crack	220/80

**Table 4 sensors-22-02579-t004:** Distribution of health states in two source domains and one target domain.

States	Source Domain Dataset *A*		Source Domain Dataset *B*		Target Domain Data	
	Data	Labels	Data	Labels	Data	Labels
1	√	√			√	
2	√	√			√	
3	√	√			√	
4	√	√	√	√	√	
5	√	√	√	√	√	
6			√	√	√	
7			√	√	√	

**Table 5 sensors-22-02579-t005:** Five different test scenarios.

Test Scenarios	Source Dataset *A*	Source Dataset *B*	Target Data
A1	Load 0% (states 1–5)	Load 20% (states 4–7)	Load 40% (states 1–7)
B1	Load 0% (states 1–5)	Load 40% (states 4–7)	Load 20% (states 1–7)
C1	Load 40% (states 1–5)	Load 20% (states 4–7)	Load 0% (states 1–7)
D1	Load 20% (states 1–5)	Load 0% (states 4–7)	Load 40% (states 1–7)
E1	Load 40% (states 1–5)	Load 0% (states 4–7)	Load 20% (states 1–7)

**Table 6 sensors-22-02579-t006:** Results of different classifiers.

Test Scenarios	Strong Classifier C*_A_*	Strong Classifier C*_B_*	Weak Classifier C*_W_*	Proposed Method
A1	92.14%	98.58%	85.89%	91.08%
B1	95.15%	98.28%	92.14%	95.41%
C1	81.50%	99.68%	78.03%	83.75%
D1	99.50%	91.07%	89.07%	92.89%
E1	98.14%	96.56%	87.50%	90.48%
Average	93.29%	96.83%	86.52%	90.73%

**Table 7 sensors-22-02579-t007:** Results of different methods.

Test Scenarios	Method 1 (CNN Trained by Source *A*)	Method 2 (CNN Trained by Source *B*)	Method 3 (DACNN Trained by Source *A*)	Method 4 (DACNN Trained by Source *B*)	The Proposed Method
A1	62.86%	55.54%	64.82%	57.14%	91.08%
B1	61.43%	56.43%	65.71%	57.28%	95.41%
C1	53.04%	54.89%	55.71%	56.42%	83.75%
D1	57.86%	53.93%	70.71%	56.25%	92.89%
E1	59.14%	55.54%	63.14%	56.96%	90.48%
Average	58.87%	55.27%	64.02%	56.79%	90.73%

**Table 8 sensors-22-02579-t008:** Four different loads.

Loads	Values
Load 1	1797 rpm, 0 hp
Load 2	1772 rpm, 1 hp
Load 3	1750 rpm, 2 hp
Load 4	1750 rpm, 3 hp

**Table 9 sensors-22-02579-t009:** Details of the test bearing.

Parameters	Values
Type	6205-2RS JEM SKF
The number of balls	9
Pitch diameter	1.537 inches
Ball diameter	0.3126 inches
Sampling frequency	12 (kHz)
Motor speed	1797/1772/1750/1730 rpm

**Table 10 sensors-22-02579-t010:** The details of the 12 operating states.

Labels	Failure Location	Failure Orientation	Failure Severities (Inches)	The Number of Testing/Training Samples
0	Health	-	0	100/200
1	Rolling element	-	0.007	100/200
2	Rolling element	-	0.014	100/200
3	Rolling element	-	0.021	100/200
4	Inner race	-	0.007	100/200
5	Inner race	-	0.021	100/200
6	Inner race	-	0.028	100/200
7	Outer race	Center	0.007	100/200
8	Outer race	Vertical	0.007	100/200
9	Outer race	Center	0.014	100/200
10	Outer race	Center	0.021	100/200
11	Outer race	Vertical	0.021	100/200

**Table 11 sensors-22-02579-t011:** Distribution of health states in source and target data.

States	Source Domain Dataset *A*		Source Domain Dataset *B*		Target Domain Data	
	Data	Labels	Data	Labels	Data	Labels
1	√	√			√	
2	√	√			√	
3	√	√			√	
4	√	√			√	
5	√	√			√	
6	√	√	√	√	√	
7	√	√	√	√	√	
8	√	√	√	√	√	
9			√	√	√	
10			√	√	√	
11			√	√	√	
12			√	√	√	

**Table 12 sensors-22-02579-t012:** Five different test scenarios.

Test Scenarios	Source Dataset *A*	Source Dataset *B*	Target Data
A2	Load 1 (states 1–8)	Load 2 (states 6–12)	Load 3 (states 1–12)
B2	Load 3 (states 1–8)	Load 4 (states 6–12)	Load 1 (states 1–12)
C2	Load 2 (states 1–8)	Load 3 (states 6–12)	Load 4 (states 1–12)
D2	Load 1 (states 1–8)	Load 2 (states 6–12)	Load 4 (states 1–12)
E2	Load 2 (states 1–8)	Load 3 (states 6–12)	Load 1 (states 1–12)

**Table 13 sensors-22-02579-t013:** Results of different methods.

Test Scenarios	Method 1 (CNN Trained Using Source Dataset *A*)	Method 2 (CNN Trained Using Source Dataset *B*)	Method 3 (DACNN Trained Using Source Dataset *A*)	Method 4 (DACNN Trained Using Source Dataset *B*)	The Proposed Method
A2	63.17%	57.13%	65.75%	58.33%	98.08%
B2	60.50%	58.25%	65.83%	58.08%	95.41%
C2	66.50%	58.08%	66.67%	58.33%	99.66%
D2	66.08%	58.14%	66.58%	58.33%	99.25%
E2	65.08%	56.08%	66.25%	57.17%	95.83%
Average	64.27%	57.53%	66.22%	58.05%	97.65%

## Data Availability

The data presented in this study are available on request from the corresponding author.

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
