# Peer review of "Partial Transfer Ensemble Learning Framework: A Method for Intelligent Diagnosis of Rotating Machinery Based on an Incomplete Source Domain"

_sensors, 2022, doi:10.3390/s22072579_

Round 1

Reviewer 1 Report

The paper presents a framework of transfer learning to diagnose faults from two incomplete datasets. The paper is structured well, and the sufficient explanations are provided where required. Few corrections/ suggestions to improve the paper are below.

  1. Partial transfer learning applied in other types of incomplete data sources in fault diagnosis (e.g. sensor data with different frequency) are not highlighted in the paper.
  2. Real-world examples of the problem statement need to be added.
  3. In pg 4, equation 8, λ is introduced without any description.
  4. In section 3.1 (pg 5), if i is always 1, x is sufficient rather than xi. If there is a case of i>1, it is not obvious in the explanation.
  5. Term ‘chapter’ is used in place of ‘section’ in some places. Consistency is required in the paper.
  6. In Eq (14) in pg 7, the third part has max(pa, pw, pw) instead of max(pa, pb, pw).
  7. Fig 7(b) in Pg 9 has numbers in the layout which is explained only in table 2 Pg 10. Putting them side by side will be easier to understand.
  8. In fig 8 (pg 9), six different faults are explained. But in print it looks like fig 8 6. Same concern with Table 3 7 on pg 10.
  9. In fig 9 and 12, mention the load at which the signals and distributions are recorded.
  10. In section 4.1.2, it is stated as ‘5 kinds of states’. It is better to mention as ‘5 kind of health states). This clarification is needed in many places in the paper.
  11. Test scenarios are denoted as A-E for both case studies. They could be differentiated as A1-E1 and A2-E2 (or similar methods to differentiate the two cases).
  12. Reference 32 is not of a proper format.
  13. List of abbreviations is required.
  14. Grammatical errors need to be checked.

Reviewer 2 Report

This paper proposed a partial transfer ensemble learning framework for the intelligent diagnosis of rotating machinery based on an incomplete source domain. This scenario is easy to occur in actual industrial engineering and this is challenging. However, there are some problems as follows:

  1. In case 2, the author should provide relevant fault pictures.
  2. For the dataset in Case, please insert a citation to it. (Version, etc.)
  3. In table 6, why the accuracy of the proposed method is lower than some of the other methods.
  4. Please check all equations, symbols, for example, formula 9 is not centred.
  5. The accuracy is about 91%, it looks not very high.
  6. Classifier Cw provides pseudo labels, the necessity should be further explained.
